# *CaCML13* Acts Positively in Pepper Immunity Against *Ralstonia solanacearum* Infection Forming Feedback Loop with CabZIP63

**DOI:** 10.3390/ijms21114186

**Published:** 2020-06-11

**Authors:** Lei Shen, Sheng Yang, Deyi Guan, Shuilin He

**Affiliations:** 1National Education Ministry Key Laboratory of Plant Genetic Improvement and Comprehensive Utilization, Fujian Agriculture and Forestry University, Fuzhou 350002, China; shorttubelycoris07@163.com (L.S.); yangsheng2061@163.com (S.Y.); gdyfujian@126.com (D.G.); 2Key Laboratory of Applied Genetics of universities in Fujian Province, Fujian Agriculture and Forestry University, Fuzhou 350002, China; 3Agricultural College, Fujian Agriculture and Forestry University, Fuzhou 350002, China

**Keywords:** *Capsicum annuum*, CaCML13, CabZIP63, immunity, *Ralstonia solanacearum*

## Abstract

Ca^2+^-signaling—which requires the presence of calcium sensors such as calmodulin (CaM) and calmodulin-like (CML) proteins—is crucial for the regulation of plant immunity against pathogen attack. However, the underlying mechanisms remain elusive, especially the roles of CMLs involved in plant immunity remains largely uninvestigated. In the present study, *CaCML13*, a calmodulin-like protein of pepper that was originally found to be upregulated by *Ralstonia solanacearum* inoculation (RSI) in RNA-seq, was functionally characterized in immunity against RSI. CaCML13 was found to target the whole epidermal cell including plasma membrane, cytoplasm and nucleus. We also confirmed that *CaCML13* was upregulated by RSI in pepper roots by quantitative real-time PCR (qRT-PCR). The silencing of *CaCML13* significantly enhanced pepper plants’ susceptibility to RSI accompanied with downregulation of immunity-related *CaPR1*, *CaNPR1*, *CaDEF1* and *CabZIP63*. In contrast, *CaCML13* transient overexpression induced clear hypersensitivity-reaction (HR)-mimicked cell death and upregulation of the tested immunity-related genes. In addition, we also revealed that the G-box-containing *CaCML13* promoter was bound by CabZIP63 and *CaCML13* was positively regulated by CabZIP63 at transcriptional level. Our data collectively indicate that CaCML13 act as a positive regulator in pepper immunity against RSI forming a positive feedback loop with *CabZIP63*.

## 1. Introduction

Plants are frequently exposed to potentially pathogenic microbes and have evolved a sophisticated defense system initiate defense upon pathogen attack. This process is largely regulated at the transcriptional levels by the action of different transcription factors, which themselves are activated by early stages of signaling. However, how the upstream defense-signalings are interconnected with transcription factors to activate appropriate transcription outputs upon attack of pathogens is not fully understood.

Subcellular fluctuations in Ca^2+^ ion concentration are among the earliest responses to pathogen attack; Ca^2+^-signaling is essential for plant immunity [1,2]. The Ca^2+^ signature is decoded and transmitted downstream by different Ca^2+^ sensors including calmodulin like (CML) proteins, calcium-dependent protein kinases (CDPKs) and calcineurin B-like proteins (CBLs). These Ca^2+^ sensors relay or decode the encoded Ca^2+^ signals into specific cellular and physiological responses in order to survive challenges by pathogens [3,4,5]. Calmodulin (CaM), which is a ubiquitous Ca^2+^ sensor in eukaryotes, plays a major role in translating these Ca^2+^ signatures to cellular responses. CaM has four Ca^2+^-binding EF-hand motifs. Upon binding to Ca^2+^, CaM undergoes a conformational change, which allows it to bind to numerous target proteins and modulate the activity of these target proteins including transcription factors involved in plant defense [6,7,8,9,10,11], as they are crucial for plant immunity, they have been found to be targeted by effectors from pathogens to repress pathogen associated molecular pattern triggered immunity (PTI) in the host plants [12,13]. Although CML proteins are structurally similar to calmodulin, the number of CMLs in plants is much higher than CaMs, for example, five CaMs and 32 CMLs in rice [14], four CaMs and 58 CMLs in apple [15]. CML8, CML9, CML24 in *Arabidopsis thaliana* act positively in immunity against *Pseudomonas syringae* pv. *tomato* DC3000 [16,17,18,19], while CML46 and CML47 in *Arabidopsis thaliana* act as negative regulators in immunity against *Pseudomonas syringae* pv *maculicola* [20]. A CML-interacting partner, PSEUDO-RESPONSE REGULATOR 2 (PRR2), a plant specific transcription factor has been found to act positively in salicylic acid (SA) and camalexin accumulation during plant immunity [21]. However, the role of CMLs in plant immunity are largely unknown.

Pepper (*Capsicum annuum*) is an agriculturally important vegetable worldwide and a Solanaceae originated from tropical and subtropical regions of Central and South America and distributed mainly in uplands in the tropical and subtropical regions with various soil-borne pathogens [22,23]. Bacterial wilt caused by infection of *R. solanacearum* is one of the most frequently occurred soil-borne diseases of pepper causing great loss in pepper production worldwide [22,24]. Our previous studies showed that CaWRKY6, CaWRKY40, CabZIP63 and CaCDPK15 act as positive regulators of pepper immunity against *R. solanacearum* infection, with *CaWRKY40* being directly regulated by CaWRKY6 and CabZIP63 and indirectly by CaCDPK15, forming positive feedback loops manner [25,26,27]. These results indicate that Ca^2+^-signaling contributes to pepper immunity against *R. solanacearum* and is closely related to CabZIP63/CaWRKY40 transcriptional cascades. In this study, we report that CaCML13, a CML protein in pepper, acts as positive regulator in pepper immunity against *R. solanacearum* forming a positive feedback loop with CabZIP63.

## 2. Results

### 2.1. The CML Gene Was Upregulated by R. solanacearum Inoculation

RNA-seq data of pepper challenged with *R. solanacearum* was analyzed. A *CML* transcript that was found to be upregulated by *R. solanacearum* infection in roots of pepper (Figure 1a). To confirm this results, we performed another experiment by quantitative real-time PCR (qRT-PCR) to assay the transcript levels of this genes upon challenge of R. solanacearum inoculation (RSI) in the roots of pepper plants, the result showed that the transcript expression levels of this *CML* gene were enhanced from 3 h post-inoculation by RSI until 48 h post-inoculation with the peak observed at 12 h post-inoculation (Figure 1b), indicating that this *CML* was upregulated by RSI.

### 2.2. The CML Is CaCML13 Exhibiting High Sequence Similarities to CMLs in Other Plant Species

The deduced amino acid sequence of the RSI upregulated pepper CML gene, encodes a 16 kDa protein with 444 bps, exhibited highest sequence similarities to CML13 among all of the CML members in different plant species including plants in the family Solanaceae, such as *Solanum lycopersicum* (*CML13*, XP_004234945.1) with 98% identity, *Solanum tuberosum* (*CML13*, XP_006340368.1) with 95% identity, *Nicotiana tabacum* (*CML13*, XP_016434602.1) with 92% identity and other plant species including *Sesamum indicum* (*CML13*, XP_011101132.1) with 91% identity, *Ipomoea nil* (*CML13*, XP_019192673.1) with 90% identity, *Jatropha curcas* (*CML13*, XP_012087102.1) with 89% identity, *Citrus sinensis* (*CML13*, XP_006477274.1) with 90% identity and *Arabidopsis thaliana* (*CML13*, NP_172695.1) with 86% identity (Figure 2a,b). Since it has the highest homology with *AtCML13* in all *CML* gene family members of *Arabidopsis thaliana*, this gene was named *CaCML13* (XP_016564798.1). Unlike CaMs that generally have four EF-hand motifs, they have only three EF-hand motifs. CaCML13 displays higher sequence similarities to its orthologs in Solanaceae than that to its orthologs in other plants including *Arabidopsis thaliana* (Figure 2).

### 2.3. CaCML13 Is a Protein Distributes in the Whole Cell Including Plasma Membrane, Cytoplasm and the Nucleus

To assay the subcellular targeting of CaCML13, we transient-overexpressing fused CaCML13-GFP protein by infiltration of *Agrobacterium tumefaciens* GV3101 cells containing *35S:CaCML13-GFP* (using *35S:GFP* as negative control). The GFP signals were detected at 48 h post-infiltration (hpi) using a confocal microscope. GFP signals within *35S:GFP*-infiltrated epidermal cells of *Nicotiana. benthamiana* leaves were observed in plasma membrane, cytoplasm and the nucleus. Similarly, the GFP signals of CaCML13-GFP-expressed cells were also visualized in plasma membrane, cytoplasm and the nucleus (Figure 3), indicating that CaCML13 distributes in the whole cells including plasma membrane, cytoplasm and the nucleus.

### 2.4. The Silencing of CaCML13 Significantly Enhanced Susceptibility to RSI

As *CaCML13* is regulated by RSI, we speculate that CaCML13 may be involved in pepper immunity against RSI. To test this possibility, we successfully and specifically silenced *CaCML13* in pepper plants by an approach of virus-induced gene silencing (VIGS), displayed with significantly lower levels of *CaCML13* (Figure 4a). The *CaCML13* silenced pepper plants were further challenged with RSI, the result showed that *CaCML13* silenced pepper plants exhibited increased susceptibility to RSI (Figure 4b), displayed enhanced wilt symptoms and larger population of *R. solanacearum* (colony-forming units, cfu) (Figure 4c), higher disease index from 5 to 14 dpi (Figure 4d). Consistently, genes encoding pathogenesis-related proteins such as *CaPR1* [28,29,30], *CaNPR1* [31,32] and *CaDEF1* [29,33] were significantly downregulated by *CaCML13* silencing (Figure 4e). These results indicate that *CaCML13* acts as positive regulator in pepper immunity against RSI.

### 2.5. Transient Overexpression of CaCML13 Triggered Intensive Hypersensitivity Reaction (HR)-Mimicked Cell Death

To confirm the results that CaCML13 acts as positive regulator in pepper immunity, we assessed the effect of *CaCML13* transient overexpression on HR-mimicked cell death and transcript expression levels of defense associated pathogenesis-related (PR) genes; the *A. tumefaciens* GV3101 cells harboring 35S:*CaCML13* or 35S:*00* (act as negative control) were infiltrated into pepper plant leaves. By analysis of qRT-PCR and western blotting with the protein isolated from *CaCML13-HA* transiently overexpressing in pepper leaves against anti-HA antibody, *CaCML13-HA* appeared to be successfully expressed in pepper leaves at transcriptional and post-translational level, respectively (Figure 5a,b). The intensive cell death was induced by infiltration of *A. tumefaciens* GV3101 harboring *35S:CaCML13-HA*, accompanied with enhanced ion leakage displayed with high level of conductivity, darker trypan blue staining and darker diaminobenzidine (DAB) staining, indicator of H_2_O_2_ accumulation, while that containing 35S:*00* did not trigger any cell death, clear trypan blue staining or DAB staining (Figure 5c–e). Consistently, the transient overexpression of *CaCML13-HA* significantly enhanced the relative transcript expression levels of *CaPR1*, *CaNPR1* and *CaDEF1* (Figure 5f). These results indicate that the transient overexpression of *CaCML13* significantly induced HR-mimicked cell death and immunity associated marker genes expression.

### 2.6. The G-Box-Containing CaCML13 Promoter Are Directly Bound by CabZIP63

The predicted region about 2000 bps length acted as the promoter of *CaCML13* was searched by NCBI website (https://www.ncbi.nlm.nih.gov/) using *CaCML13* ORF sequence. A G-box *cis*-element was found in the promoter of *CaCML13* (Figure 6a). This box was previously found to be bound by bZIP transcription factor [34,35]; CabZIP63 was previously found by us to be involved in pepper immunity against RSI via targeting G-box-containing-immunity-related genes [26]. We speculate that CaCML13 may be directly targeted by CabZIP63. To confirm this hypothesis, we performed a chromatin immunoprecipitation (ChIP)-PCR to test the direct binding of CabZIP63 to G-box-containing promoter of *CaCML13*. The leaves of pepper plants were infiltrated with *A. tumefaciens* GV3101 cells containing *35S:CaCML13-HA*, which were harvested at 48 hpi for formaldehyde cross-linking, chromatins isolation, fragmentation, immunoprecipitation with antibodies of HA and DNA purification and PCR with appropriate primer pair. The result showed *CaCML13-HA* was successfully expressed (Figure 6b) and a clear enrichment of CabZIP63 was found on the G-box-containing promoter fragment of *CaCML13*, but not in the G-box-free control promoter fragment of *CaCML13* (Figure 6c). This result indicate that CabZIP63 directly targets *CaCML13*.

### 2.7. The Interrelationship between CabZIP63 and CaCML13 at the Transcriptional Level

The binding of G-box-containing *CaCML13* promoter fragments by CabZIP63 indicates that *CaCML13* may be regulated by CabZIP63 at the transcriptional level. To test this possibility, the relative transcript expression level of *CaCML13* upon transient overexpression of *CabZIP63* in pepper leaves was detected by qRT-PCR analysis. The success of *CabZIP63* expression was confirmed, and the results showed that the relative transcript expression levels of *CaCML13* were significantly enhanced by *CabZIP63* transient overexpression (Figure 7a). In contrast, the relative transcript levels of *CaCML13* were detected in the control and *CabZIP63*-silenced pepper leaves inoculated with *R. solanacearum* by qRT-PCR, the results showed that *CabZIP63* was successful silenced and the relative transcript levels of *CaCML13* were significantly downregulated by this silencing (Figure 7b). On the other hand, to investigate whether the transcript levels of *CabZIP63* is regulated by CaCML13, the relative transcript levels of *CabZIP63* were detected upon transient overexpression of *CaCML13* or its silencing in leaves of pepper plant inoculated with *R. solanacearum*. The results showed that the relative transcript levels of *CabZIP63* were significantly upregulated by transient overexpression of *CaCML13*, but downregulated by *CaCML13* silencing (Figure 7c,d). These data suggest that *CabZIP63* regulates *CaCML13* transcript expression level by the targeting to *CaCML13* promoter and formed a positive feedback loop with *CaCML13* at transcriptional level.

## 3. Discussion

Ca^2+^-signaling is crucial for the regulation of plant immunity against pathogen attack, the underlying mechanism remain elusive, especially the role of CMLs in plant immunity remains largely uninvestigated. In the present study, we uncover that *CaCML13* is upregulated by attack of *R. solanacearum* attack and acts a positive regulator in pepper immunity against *R. solanacearum*.

A key step in both PTI and ETI is the upregulation of a multitude of defense genes, previous studies suggest that genes upregulated in host plants upon pathogen infection generally have important roles for disease resistance [25,26,27,36,37,38]. As *CaCML13* was upregulated by RSI, implying its possible role in pepper immunity against *R. solanacearum* infection. This speculation was confirmed by the results that silencing of *CaCML13* by VIGS significantly enhanced susceptibility of pepper plants to RSI, coupled with downregulation of immunity-related *CaPR1* [28,29,30] and *CaNPR1* [31,32], *CaDEF1* [29,33] and *CabZIP63* [26]. In contrast, through the transient overexpression, which were frequently employed previously [39,40,41], we found that CaCML13 acts positively in pepper immunity manifested by clear HR-mimicked cell death displayed enhanced ion leakage and trypan blue staining as well as H_2_O_2_ accumulation displayed by darker DAB staining. Since H_2_O_2_ accumulation was closely related to HR cell death [42,43], which are generally believed to be accompanied with effector triggered immunity [44]. All these data indicate that CaCML13 acts as a positive regulator in pepper immunity against *R. solanacearum* infection. Taken together, we propose a working model that illustrate the positive feedback loop between *CaCML13* and *CabZIP63* in pepper response to *R. solanacearum* infection (Figure 8). *CaCML13*, induced by *R. solanacearum* infection, was targeted by CabZIP63 and formed a positive feedback loop with *CabZIP63* at the transcriptional level in pepper response to *R. solanacearum* infection. 

Noticeably, CabZIP63 act as positive regulator in pepper response to *R. solanacearum* infection by upregulating PR genes including *CaPR1*, *CaNPR1*, *CaDEF1* and the data in the present study showed that CaCML13 positively regulates these genes, indicating that CaCML13 is related to CabZIP63 in terms of both expression and function. This speculation was further confirmed by the data that *CaCML13* is directly targeted by CabZIP63 probably in G-box dependent manner and is positively regulated by CabZIP63. Given that *CabZIP63* was upregulated by transient overexpression of *CaCML13* and downregulated by *CaCML13* silencing, it can be concluded that a positive feedback loop between *CaCML13* and *CabZIP63*. Similar positive feedback loops were observed in the same pathosystem between *CabZIP63* and *CaWRKY40* [26], *CaCDPK15* and *CaWRKY40* [27], *CaCBL1* and *CaWRKY40* [45] and in other pathosystems [46,47], these positive feedback loops may be essential for immune signal amplification during plant response to pathogens. The close relationship between G-box dependent function of *CabZIP63* and *CaCML13* also implies the possible involvement of *CaCML13* in pepper response to abiotic stresses such as heat stress and its association to ABA-signaling, as *CabZIP63* has been previously found to be related to abscisic acid (ABA)-signaling-dependent *CaWRKY40* and act as positive regulator in thermotolerance [26] and G-box was believed to be a class of *cis*-acting elements related to ABA-signaling and bZIPs [48,49,50], to confirm this speculation, further investigation is required. In addition. our data also indicate that upon the activation of Ca^2+^ influx, multiple Ca^2+^ sensors may participate in the coordinated transmission of Ca^2+^-signaling, since CaCDPK15, CaCBL1 and CaCML13 are all participate in the pepper defense-signaling against *R. solanacearum* infection. This arrangement may make the immune-signaling less sensitive to the attacks of pathogen derived effectors, once per component is destroyed by pathogens, it may be compensated on time by other components, since components in Ca^2+^-signaling such as calmodulin may be target of pathogen derived effectors [12,13]. Another biologic importance of the involvement of multiple-signaling components in defense-signaling is that it can provide great regulatory potential for the feasible response of plants to different environmental conditions, since different signaling components can be modulated by internal or external stimulus.

Collectively, our data indicate that CaCML13 acts as positive regulator in pepper immunity against *R. solanacearum* infection forming a positive feedback loop with CabZIP63. Our findings imply that Ca^2+^-signaling mediated by multiple Ca^2+^ sensors including CaCML13, ABA-signaling and transcriptional cascades including CabZIP63 and CaWRKY40 are involved in pepper response to *R. solanacearum* infection and probable in thermotolerance, provide start points for further elucidation of mechanism underlying the crosstalk between pepper response to *R. solanacearum* infection and high temperature and high humidity in the future.

## 4. Materials and Methods

### 4.1. Plant Materials and Growth Conditions

The pepper inbred line HN42 and *N. benthamiana* plants were grown in sterilized soil in square plastic pots (7 × 7 cm) in a growth room at 25 °C and 60% humidity, with a light intensity of 60–70 μmol photons m^−2^ s^−1^ in a 16-h light/8-h dark photoperiod.

### 4.2. The Vectors Construction

To construct vectors for overexpression, the full-length open reading frame (ORF) of *CaCML13* or *CabZIP63* were cloned into the entry vector pDONR207 by BP reaction, after confirmation by sequencing, they were further cloned into destination vectors pMDC83 and pEarleyGate201 by LR reaction, using Gateway cloning techniques (Invitrogen, Carlsbad, CA, USA). To construct vectors for gene silencing by VIGS, a specific 260 bp fragment in the ORF of *CaCML13*, whose sequence specificity was confirmed by BLAST searching against genome sequence in the databases of Zunla-1 (http://peppersequence.genomics.cn/page/species/blast.jsp), were amplified by PCR with the specific primer pairs using DNA of Zunla-1 as template, and then cloned into pDONR207 by BP reaction after confirmation by sequencing, and then were further cloned into the PYL279 (TRV2) vector. All of the vectors were transformed into *A. tumefaciens* GV3101.

### 4.3. VIGS Assay

*A. tumefaciens* GV3101 cells harboring TRV1, TRV2:*00*, TRV2:*CaPDS*, TRV2:*CaCML13* or TRV2:*CabZIP63* was cultured overnight in LB media supplemented with appropriate antibiotics, then spun down and resuspended to a concentration of OD_600_ = 0.8 in the infiltration medium (10-mM MES, 10-mM MgCl_2_, 200-mM acetosyringone, pH = 5.4). The *A. tumefaciens* GV3101 cells containing TRV1 were mixed with cells containing TRV2:*00*, TRV2:*CaPDS* or TRV2:*CaCML13* or TRV2:*CabZIP63* at a 1:1 ratio and infiltrated into the cotyledons of two-week-old pepper seedlings. The plants were then placed in a growth chamber at 16 °C in the dark for 56 h, and then transferred to a growth room at 25 °C and 60% humidity, with a light intensity of 60–70 μmol photons m^−2^ s^−1^ and a 16-h light/8-h dark photoperiod, until the TRV:*CaPDS* plants exhibited a bleached phenotype.

### 4.4. Transient Overexpression of CaCML13-HA in Pepper Leaves

For transient expression analysis, *A. tumefaciens* GV3101 cells containing *35S:CaCML13-HA* (using *35S:HA* as control) were grown overnight and then resuspended in induction medium to OD_600_ = 0.8, approximately 100 μL was infiltrated into one infiltrated site in the pepper plant leaves at the eight-leaf stage by using a syringe without a needle. The infiltrated leaves were harvested at the indicated time points for further use.

### 4.5. R. solanacearum Growth and RSI

Pepper plants or pepper leaves were inoculated with the *R. solanacearum* strain FJC100301 as described previously (Shen et al., 2016a; Qiu et al., 2018). *R. solanacearum* was grown in a SPA liquid medium (200-g/L potato extract, 20-g/L sucrose, 3-g/L beef extract, 5-g/L tryptone) at 28 °C, then centrifuged and resuspended in a 10-mM MgCl_2_ solution to a concentration of 10^8^ cfu/mL. The cells were used to inoculate pepper plants via root irrigation or pepper leaves by injection.

### 4.6. Subcellular Localization

The subcellular localization assay was performed as previously described (Shen et al., 2016a). The *A. tumefaciens* GV3101 cells containing *35S:CaCML13-GFP* were infiltrated into *N. benthamiana* leaves, and the GFP signals were observed at 48 hpi under a laser scanning confocal microscope (TCS SP8; Leica Microsystems, Weztlar, Germany).

### 4.7. ChIP Analysis

A ChIP assay was performed as previously described [51,52]. Briefly, pepper leaves infiltrated with *A. tumefaciens* GV3101 cells containing *35S:HA* or *35S:CabZIP63-HA* were harvested at 48 hpi for chromatin isolation. The extracted chromatins were sonicated to generate DNA fragments between 200 and 500 bp in length, which were incubated with magnetic beads (Cat#88803, Thermo Fisher Scientific, Waltham, Massachusetts, USA) linked with anti-HA antibody for 2 h at 4 °C, following the manufacturer’s instructions. The immunoprecipitated DNA fragments were used as a template to analyze the enrichment of CabZIP63 on G-box-containing promoters of *CaCML13* by using PCR with the specific primers listed in Appendix A.

### 4.8. qRT-PCR Assay

To detect the transcript expression levels of the selected genes, the qRT-PCR assay was performed by using a Bio-Rad Real-Time PCR system (Bio-Rad Laboratories, California, USA) and the SYBR Premix Ex Taq II system (Takara Bio, Kyoto, Japan) with the specific primers listed in Appendix A. The expression level of *CaActin* (GQ339766) was monitored as an internal reference gene to normalize the transcript expression levels. The Livak method was used to analyze the data [53].

### 4.9. Measurement of Ion Conductivity

Ion leakage was measured following a previously described method [26,54,55]. Leaf disks (6 mm in diameter) were taken from pepper leaves infiltrated with *A. tumefaciens* GV3101 cells harboring *35S:HA* or *35S:CaCML13-HA*.The disks were harvested at different time points and then incubated in 5 mL of double distilled water (ddH_2_O) for 1 h at 28 °C. The ion conductivity was measured using a Mettler Toledo 326 ion meter (Mettler Toledo, Zurich, Switzerland).

### 4.10. Histochemical Staining Assay

To assess the HR-mimicked cell death by transient overexpression of *CaCML13* in pepper leaves, histochemical staining assays including trypan blue staining and DAB staining were performed as previously described [26,27,39].

### 4.11. Immunoblot Analysis

The expression of *CaCML13* or *CabZIP63* at the post-transcriptional level was detected by western blot assay as described previous studies [45]. The pepper plant leaves infiltrated with *A. tumefaciens* GV3101 containing *35S:HA 35S:CaCML13-HA* or *35S:CabZIP63-HA* constructs were harvested in liquid nitrogen and ground into power. The total proteins were extracted by plant protein extraction buffer (25-mM Tris-HCl pH 7.5, 150-mM NaCl, 1-mM EDTA, 10% glycerol, 1% Triton X-100, 10-mM DTT, 1×complete protease inhibitor cocktail (Roche, Basel, Switzerland) and 2% (*w*/*v*) polyvinyl polypyrrolidone), and then incubated in ice for 1 h. The total proteins were separated by SDS-PAGE and transmitted into polyvinylidene fluoride (PVDF) membrane at 200-mA constant current for 30 min by semi-dry rotary system (Bio-Rad, California, USA). The PVDF membrane was soaked in blocking buffer for 1 h at RT and then incubated with primary anti-HA-tag mAb antibody (Cat#M180-3, MBL, Tokyo, Japan) at a 1:5000 dilution. Anti-IgG (H+L chain) (Mouse) pAb-HRP (Cat#458, MBL, Tokyo, Japan) was used as a secondary antibody, diluted at 1:20,000.

### 4.12. Calculation of Disease Index of Bacterial Wilt

Twenty *R. solanacearum*-inoculated pepper plants were scored every day using a disease index ranging from 0 to 4:0 (no wilting), 1 (1–25% wilted), 2 (26–50% wilted), 3 (51–75% wilted) and 4 (76–100% wilted or dead).

## Figures and Tables

**Figure 1 ijms-21-04186-f001:**
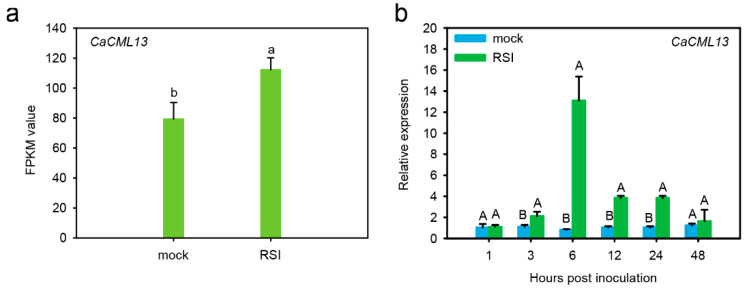
Transcriptional response of *CaCML13* to RSI. (**a**) fragments per kilobase million (FPKM) of *CaCML13* in pepper roots by RNA-seq. The RNA-seq data used in this study have been deposited at the China National GeneBank DataBase (CNP0001104); (**b**) relative transcript levels of *CaCML13* in roots of pepper plants challenged with RSI by qRT-PCR. Data presented are means ± standard error of four replicates. Different uppercase letters above the bars indicate significant differences among means (*p* < 0.01) by Fisher’s protected least significant difference (LSD) test. Different lowcase letters above the bars indicate significant differences among means (*p* < 0.05) by Fisher’s protected LSD test.

**Figure 2 ijms-21-04186-f002:**
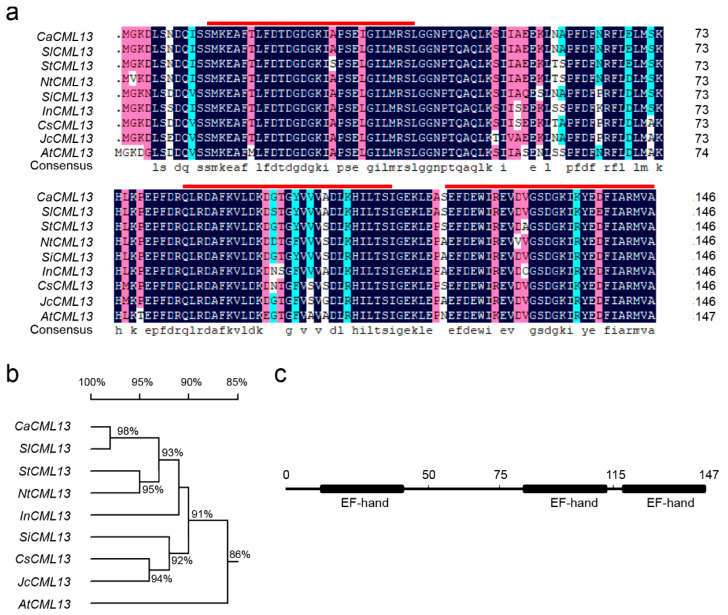
Deduced amino acid sequence of pepper *CaCML13* and its sequence similarities to its orthologs in other plant species. (**a**) Comparison of deduced amino acid sequences of pepper *CaCML13* to its orthologs in other plant species, amino acid residues that are conserved in at least five of the seven sequences are shaded, whereas amino acids identical in all seven proteins are shown in black, alignments were made in DNAMAN 7 (Lynnon Biosoft, USA) using the default parameters; (**b**) phylogenetic analysis of CaCML13 with its orthologs in other plant species including *S. lycopersicum*, *S. tuberosum*, *N. tabacum*, *S. indicum*, *I. nil*, *J. curcas*, *C. sinensis* and *A. thaliana*; (**c**) highly conserved EF-hand motifs in CaCML13.

**Figure 3 ijms-21-04186-f003:**
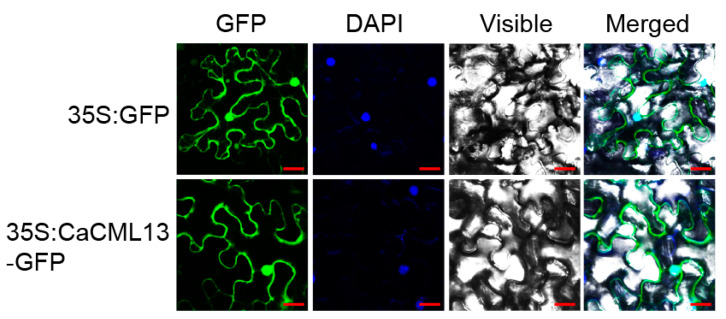
Subcellular localization of CaCML13 in *N. benthamiana* epidermal cells. *N. benthamiana* leaves were infiltrated with *A. tumefaciens* GV3101 cells harboring *35S:CaCML13-GFP* (using *35S:GFP* as control). Subcellular localization of the CaCML13-GFP fusion protein or GFP was viewed with a laser scanning confocal microscope at 48 hpi. The nucleus was displayed by diamidine phenyl indole (DAPI) staining, fluorescence images (GFP), bright-field images (Visible), and the corresponding overlay images (Merged) of representative cells expressing GFP or CaCML13-GFP fusion protein are shown. Bars = 25 µm.

**Figure 4 ijms-21-04186-f004:**
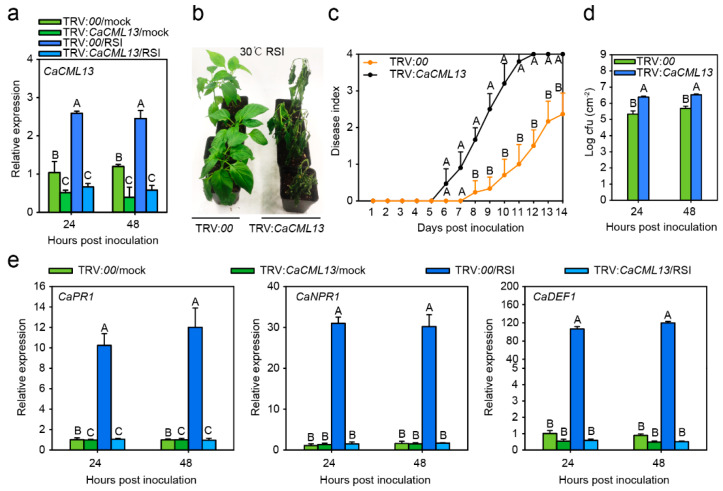
*CaCML13*-silencing-impaired pepper immunity against RSI coupled with downregulation of PR genes. (**a**) Test of the success of *CaCML13* silencing in TRV:*CaCML13* pepper plant leaves challenged with RSI at the transcriptional level at 24 h post-inoculation; (**b**) bacterial wilt symptoms in the *CaCML13* silenced and the control pepper plants at 11 days post inoculation by root-irrigation; (**c**) disease index from 5 to 14 dpi in *CaCML13* silenced pepper and the control plants challenged with RSI; (**d**) growth of *R. solanacearum* displayed with cfu in *CaCML13* silenced and the control pepper plant leaves challenged with RSI at 24 and 48 h post-inoculation; (**e**) relative transcript levels of immunity associated genes including *CaPR1*, *CaNPR1* and *CaDEF1* were significantly downregulated by *CaCML13* silencing at 24 h post-inoculation. In (**a**,**c**–**e**), data presented are means ± standard error of four replicates. Different uppercase letters above the bars indicate significant differences among means (*p* < 0.01) by Fisher’s protected LSD test.

**Figure 5 ijms-21-04186-f005:**
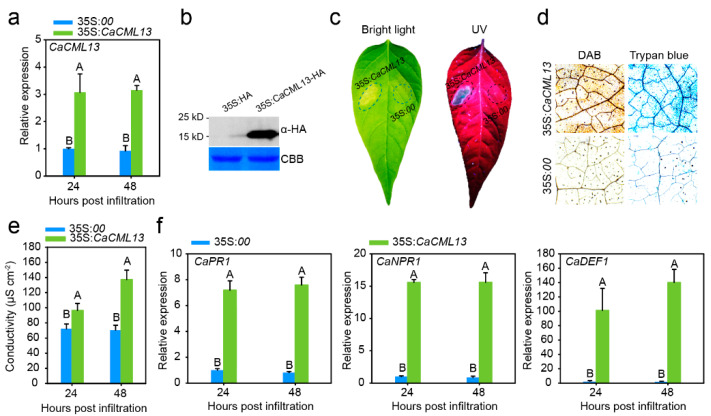
Effect of transient overexpression of *CaCML13* on HR-mimicked cell death and expression of the tested immunity-related marker genes. (**a**,**b**) success of *CaCML13* by qRT-PCR (**a**) and western blotting with antibody of HA (**b**); (**c**–**e**) HR-mimicked cell death induced by transient overexpression of *CaCML13* in pepper leaves (**c**), HR-mimicked cell death displayed with darker trypan-blue staining and H_2_O_2_-accumulation manifested by darker DAB staining and (**d**) enhanced ion leakage displayed by high level of conductivity (**e**); (**f**) effect of *CaCML13* transient overexpression on transcript levels of *CaPR1*, *CaNPR1* and *CaDEF1*. In (**a**,**e**,**f**), data presented are means ± standard error of four replicates. Different uppercase letters above the bars indicate significant differences among means (*p* < 0.01) by Fisher’s protected LSD test.

**Figure 6 ijms-21-04186-f006:**
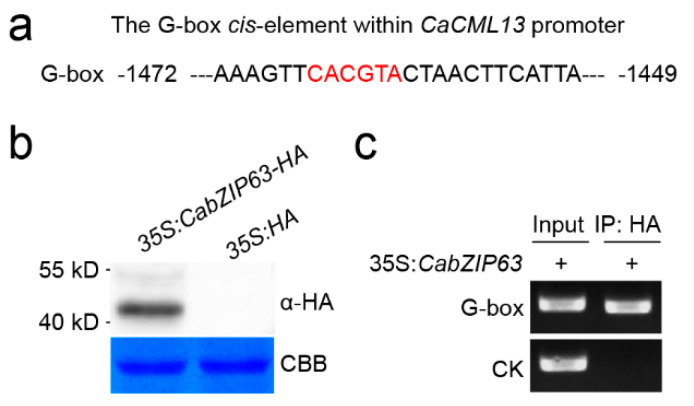
Enrichments of CabZIP63 on the promoter of *CaCML13* by ChIP-PCR. (**a**) G-box-containing promoter of *CaCML13* from −1472 to −1449 bp; (**b**) success of *CabZIP63-HA* expression in pepper leaves infiltrated with *A. tumefaciens* GV30101 cells containing *35S:CabZIP63-HA* by western blotting using antibodies of HA; (**c**) enrichments of CabZIP63 on G-box within *CaCML13* promoter by ChIP-PCR. CK that used as a negative control represents the G-box-free DNA fragment within *CaCML13* promoter. The specific primer pairs used to amplify the G-box or CK were listed in Appendix A.

**Figure 7 ijms-21-04186-f007:**
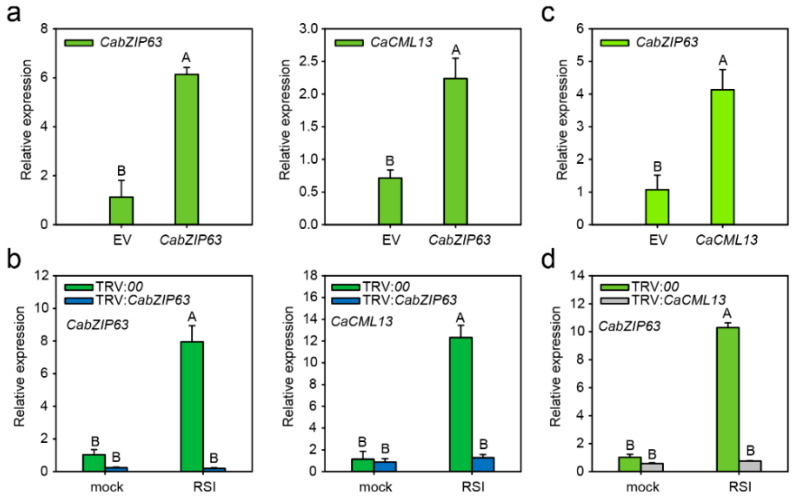
*CabZIP63* and *CaCML13* positively regulate each other at transcriptional level. (**a**) The relative transcript levels of *CabZIP63* and *CaCML13* were detected by qRT-PCR in pepper leaves upon *CabZIP63* transient overexpression at 24 hpi; (**b**) The relative transcript levels of *CabZIP63* and *CaCML13* were detected by qRT-PCR in the control and *CabZIP63*-silenced pepper plant leaves inoculated with *R. solanacearum* at 24 h post-inoculation; (**c**) The relative transcript levels of *CabZIP63* were detected by qRT-PCR upon transient overexpression of *CaCML13* in pepper leaves at 24 hpi; (**d**) The relative transcript levels of *CabZIP63* were detected by qRT-PCR in the control and leaves of *CaCML13*-silenced pepper plants inoculated with *R. solanacearum* at 24 h post-inoculation. Data presented are means ± standard error of four replicates. Different uppercase letters above the bars indicate significant differences among means (*p* < 0.01) by Fisher’s protected LSD test.

**Figure 8 ijms-21-04186-f008:**
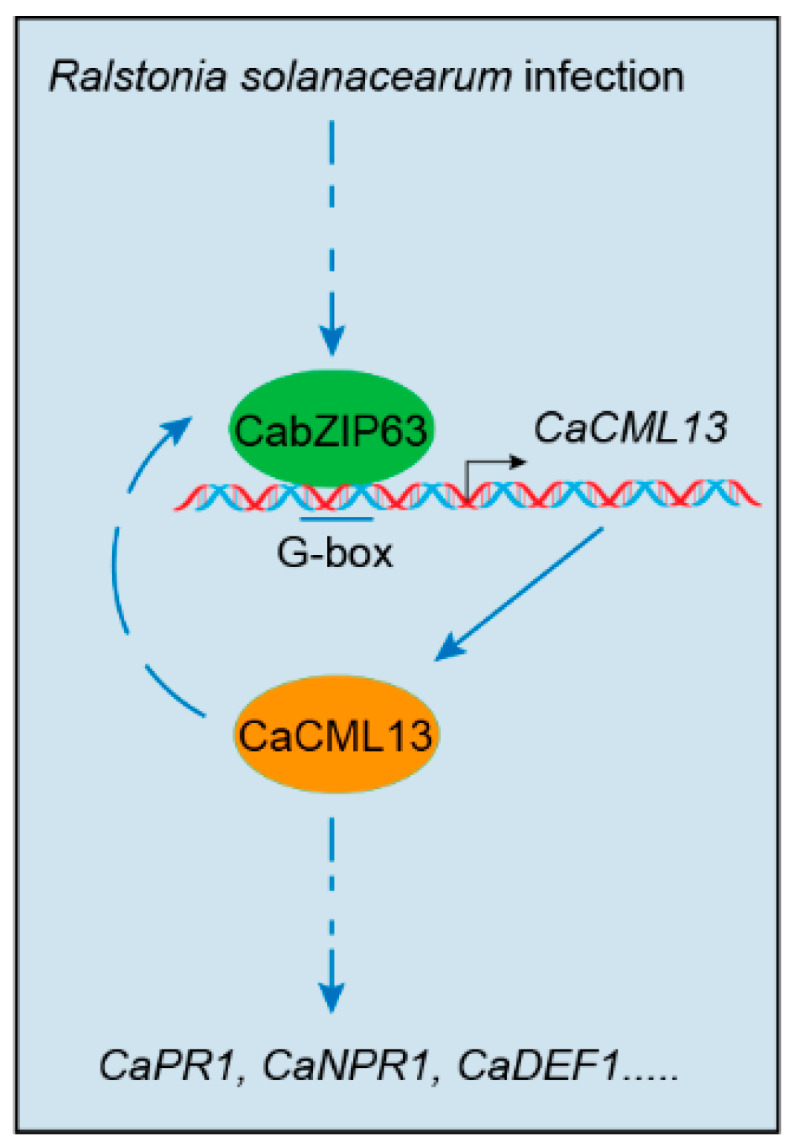
Proposed model of the positive feedback loop between *CaCML13* and *CabZIP63* in pepper response to *R. solanacearum* infection. The blue dotted arrows indicate the indirect process and the blue solid arrow indicates the direct process.

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
