# Peer review of "CaCML13 Acts Positively in Pepper Immunity Against Ralstonia solanacearum Infection Forming Feedback Loop with CabZIP63"

_ijms, 2020, doi:10.3390/ijms21114186_

Round 1

Reviewer 1 Report

The Ms describes the calmodulin like protein found in Capsicum annuum in a RNA-seq assay. The results presented are quite interesting, but the presentation has to be deeply improved.

The RNA-seq data must to be in a repository database.

There are many repetitions and there are some references that not correspond to the stated concept. 

line 48 PTI, the abbreviation needs to be explained

line 51. seems that the sentence is incomplete

line 52. the reference 20 does not correspond to the stated concept 

line 52 PRR2, the abbreviation needs to be explained

For the species I recommend not use abbreviation, the best is use the Latin binomial name 

Line 172 it is not clear how was identify the promoter of CaCML13, please explain better

I did a partial adjustment of the three first pages of the Ms that I am herein including, considering some suggestions 

Author Response

Point 1: The RNA-seq data must to be in a repository database.

Response: Thank you for your consideration of our work. The RNA-seq assay has been finished by another Ph D student in our group, in the present study we only used the data of CaCML13. If we upload the RNA-seq data to the repository database, it will affect his paper publication recently. We added “(data not shown)” in line 74 in the revised manuscript.

Point 2: There are many repetitions and there are some references that not correspond to the stated concept.

Response:Thank you for your consideration. We replaced the unsuitable references, please see the new references 20, 28, 29, 30 in lines 54, 138-139, and 239-240 in the revised manuscript, respectively.

Point 3: line 48 PTI, the abbreviation needs to be explained

Response: Thank you for your consideration. We explained the abbreviation of PTI to “pathogen associated molecular pattern triggered immunity (PTI)”, please see in line 49 in the revised manuscript.

Point 4: line 51. seems that the sentence is incomplete

Response: Thank you for your consideration. We corrected this sentence, please see in line 51 in the revised manuscript.

Point 5: line 52. the reference 20 does not correspond to the stated concept 

Response: Thank you for your consideration. We corrected the reference, please see in line 54 in the revised manuscript.

Point 6: line 52 PRR2, the abbreviation needs to be explained

Response: Thank you for your consideration. We explained the abbreviation of PRR2 to “PSEUDO-RESPONSE REGULATOR 2 (PRR2)”, please see in line 55 in the revised manuscript.

Point 7: For the species I recommend not use abbreviation, the best is use the Latin binomial name 

Response: Thank you for your suggestion. We replaced the abbreviation of species name to the Latin binomial name, please see in lines 53, 54, 65, 68, 71, 73, 75, 123, 124,136-137, 147, 155-156, 160, 187, 196, 208, 213, 223, 226, 232-233, 237, 246, 249, 271, 280, 288, 301, 303, 306, 313, 318-320, 326-327, 331, 346, 357, 368 in the revised manuscript.

Point 8: Line 172 it is not clear how was identify the promoter of CaCML13, please explain better

Response: Thank you for your consideration. We explained the identity of the CaCML13 promoter, please see in line 179-180 in the revised manuscript.

Point 9: I did a partial adjustment of the three first pages of the Ms that I am herein including, considering some suggestions 

Response: Thank you for your suggestion. We corrected the MS according to your suggestions, please see in lines 33-36, 47-55, and 59-60 in the revised manuscript.

Reviewer 2 Report

The manuscript "CaCML13 acts positively in pepper immunity against Ralstonia solanacearum infection forming feedback loop with CabZIP63" by Shen et al is a very nice study. In my opinion, the study is of great interest, however, some improvements are recommended in order to increase it's scientific merit and contribution to the subject:

  1. Authors need to improve the quality of presentations of figures.
  2.  Discussion can be improved in a more precise manner.
  3. it would be more clear if authors could make a summary diagram for the whole study indicating the mechanism of CaCML13.

Author Response

Point 1: Authors need to improve the quality of presentations of figures.

Response: Thank you for your consideration. We improved the quality of presentations of figures, please see the figures in the revised manuscript.

Point 2: Discussion can be improved in a more precise manner.

Response: Thank you for your suggestion. We improved the discussion, please see in lines 245-284 in the revised manuscript.

Point 3: it would be more clear if authors could make a summary diagram for the whole study indicating the mechanism of CaCML13.

Response: Thank you for your good suggestion. We added the summary diagram, please see Figure 8 and the figure legend of Figure 8 in lines 527-530 in the revised manuscript.

Reviewer 3 Report

Shen et al manuscript reports on the potential role of a calmodulin-like gene from pepper identified thanks to RNAseq results analysis, and proposes its characterization in planta using various approaches including transgenesis (VIGS and transient over-expression), molecular biology (including ChIP PCR) or phytopathology approaches. In particular its interconnexion with bZIP63 is proposed. Altogether the obtained results led the Authors to propose a role of CML13 in pepper resistance to the bacterial wilt caused by infection of Ralstonia solanacearum.

This research is solid and the results provide a comprehensive characterization of CaCML13 and its role in pepper immunity against Ralstonia solanacearum infection.

Results are well described, however, in my opinion Discussion is too short considering the high amount of results presented by the Authors.

In particular, Gbox is a class of cis-acting elements also known to be related to abscisic acid signaling. Considering the well-described role of ABA in plant defense response to biotic and abiotic stress, it could be relevant to include a short discussion about this point.

Future fundamental as well as applicative prospects about the present research are also awaited.

Minor points:

From Line 55 to 67: police size is different

Line 96: Solanaceae without « s »

Figure 2b: precise the bootstrapping values in figure legend

Author Response

Point 1: Results are well described, however, in my opinion Discussion is too short considering the high amount of results presented by the Authors.

Response: Thank you for your consideration. We improved the discussion, please see in lines 526-529 in the revised manuscript.

Point 2: In particular, G-box is a class of cis-acting elements also known to be related to abscisic acid signaling. Considering the well-described role of ABA in plant defense response to biotic and abiotic stress, it could be relevant to include a short discussion about this point.

Response: Thank you for your good suggestion, we added a short discussion about the possible relationship between CaCML13, G-box, bZIPs and ABA signaling in pepper immunity and thermotolerance in the Discussion section. Please see in lines 259-266 in the revised manuscript.

Point 3: Future fundamental as well as applicative prospects about the present research are also awaited.

Response: Thank you very much for your good suggestion, we added a short paragraph in the conclusion section to state the future fundamental of you findings. Please see in lines 279-284 in the revised manuscript.

Minor points:

Point 4: From Line 55 to 67: police size is different

Response: Thank you for your consideration of our paper, actually we did not know the exact meaning by " police size is different ", but we carefully checked this paragraph and improved the language. Please see in lines 58-70 in the revised manuscript.

Point 5: Line 96: Solanaceae without « s »

Response: Thank you for your consideration. We corrected it, please see in line 100 in the revised manuscript.

Point 6: Figure 2b: precise the bootstrapping values in figure legend

Response: Thank you for your consideration. We rewrote the figure legend of Figure 2b, please see in lines 108-110 in the revised manuscript.

Round 2

Reviewer 1 Report

Probably I was not clear about the scientific names writing, please follow this recommendation:

The Latin scientific name of a species is a two-part name consisting of the genus name first, and the species name second, for example Nicotiana benthamiana. After the first use, the genus name can be abbreviated to just its initial that has to be capitalized: N. benthamiana. The whole name has to be italicized. Thereafter the scientific name is written abbreviated

As far as I know, all the data and materials of a publication must be publicly available. If there is a manuscript in preparation, which is what I suppose is your case (that of the PhD student you mention), the data can be deposited and only made public until after the publication of the main article.

Please see instruction for authors:

Accession numbers of RNA, DNA and protein sequences used in the manuscript should be provided in the Materials and Methods section. Also see the section on Deposition of Sequences and of Expression Data.

Author Response

Point 1: The Latin scientific name of a species is a two-part name consisting of the genus name first, and the species name second, for example Nicotiana benthamiana. After the first use, the genus name can be abbreviated to just its initial that has to be capitalized: N. benthamiana. The whole name has to be italicized. Thereafter the scientific name is written abbreviated

Response:Thank you for your consideration. We changed the Latin scientific name of the species including Nicotiana benthamiana, Ralstonia solanacearum and Agrobacterium tumefaciens GV3101 according to your suggestion. Please see in lines 67, 69, 74, 76, 122, 123, 135, 146, 154-155, 159, 186, 195, 107, 212, 222, 225, 232 in the revised manuscript.

Point 2: As far as I know, all the data and materials of a publication must be publicly available. If there is a manuscript in preparation, which is what I suppose is your case (that of the PhD student you mention), the data can be deposited and only made public until after the publication of the main article.

Response:Thanks for your suggestion. We replaced the CaCML13 FPKM value of RNA-seq data from PhD students in our lab with another new RNA-seq data of pepper plant root inoculated with R. solanacearum. We changed the Figure 1a, please see the new Figure 1a in the revised manuscript. The new RNA-seq data were uploaded into the China National Database (https://www.cngb.org/index.html) and data will be reviewed within two days. The submission ID is CNP0001104.

Reviewer 2 Report

Authors have improved the manuscript. Authors added the summary figure as Figure 8. but I could only see the legend, not the Figure. authors forgot to add the actual Figure? 

This summary figure should be part of discussion. Please remove the Figure legend from Line 527-530 and insert it at Line 247.

Author Response

Point 1: Authors have improved the manuscript. Authors added the summary figure as Figure 8. but I could only see the legend, not the Figure. authors forgot to add the actual Figure? 

Response: Thank you for your consideration. We added the Figure 8, please see it in line 531 in the revised manuscript.

Point 2: This summary figure should be part of discussion. Please remove the Figure legend from Line 527-530 and insert it at Line 247.

Response: Thank you for your suggestion. We removed the Figure legend from line 527-530 and insert it at Line 247, please see the Figure legend in line 247 in the revised manuscript.